# Recognition of Student Engagement State in a Classroom Environment Using Deep and Efficient Transfer Learning Algorithm

**Sana Ikram [1], Haseeb Ahmad [1], Nasir Mahmood [1], C. M. Nadeem Faisal [1], Qaisar Abbas [2,\*], Imran Qureshi [2] and Ayyaz Hussain [3]**

1   Department of Computer Science, National Textile University, Faisalabad 37610, Pakistan;
    sana.imram@ntu.edu.sa (S.I.); haseeb_ad@ntu.edu.pk (H.A.); nadeem.faisal@ntu.edu.pk (C.M.N.F.)
2   College of Computer and Information Sciences, Imam Mohammad Ibn Saud Islamic University (IMSIU),
    Riyadh 11432, Saudi Arabia; iqureshi@imamu.edu.sa
3   Department of Computer Science, Quaid-i-Azam University, Islamabad 44000, Pakistan;
    ayyaz.hussain@qau.edu.pk
\*   Correspondence: qaabbas@imamu.edu.sa; Tel.: +966-537014011

**Abstract:** A student's engagement in a real classroom environment usually varies with respect to time. Moreover, both genders may also engage differently during lecture procession. Previous research measures students' engagement either from the assessment outcome or by observing their gestures in online or real but controlled classroom environments with limited students. However, most works either manually assess the engagement level in online class environments or use limited features for automatic computation. Moreover, the demographic impact on students' engagement in the real classroom environment is limited and needs further exploration. This work is intended to compute student engagement in a real but least controlled classroom environment with 45 students. More precisely, the main contributions of this work are twofold. First, we proposed an efficient transfer-learning-based VGG16 model with extended layer, and fine-tuned hyperparameters to compute the students' engagement level in a real classroom environment. Overall, 90% accuracy and 0.5 N seconds computational time were achieved in terms of computation for engaged and non-engaged students. Subsequently, we incorporated inferential statistics to measure the impact of time while performing 14 experiments. We performed six experiments for gender impact on students' engagement. Overall, inferential analysis reveals the positive impact of time and gender on students' engagement levels in a real classroom environment. The comparisons were also performed by various transfer learning algorithms. The proposed work may help to improve the quality of educational content delivery and decision making for educational institutions.

**Keywords:** student engagement; smart decision making; affective state; computer vision; deep learning; transfer learning; VGG16; class environment; demographic analysis

## 1. Introduction

Student engagement (SE) is an imperative state of learning that has been discussed since the late 1980s. Researchers compute SE while considering diverse features. SE is an integral part of the education system. SE refers to the focused interaction between educational stakeholders (precisely, instructors and students) with a goal of knowledge delivery [1]. Academic outcomes including grades may be improved if students remain engaged during the learning phase [2]. SE is also considered as an indicator to check whether active learning sessions are taking place during lecture deliverance. Researchers have the consensus that SE could be an imperative factor for achieving better outcomes, especially in higher education [3]. Around 44 studies categorize three types of engagement for learning including (1) emotional engagement, (2) behavioral engagement, and

(3) cognitive engagement. Emotional engagement is assessed through emotional reactions such as anger, disgust, fear, sadness, enjoyment, and surprise and the components for estimating behavioral engagement include effort, persistence, and attention. Cognitive engagement identifies how learners set their plans and goals and make efforts to organize their studies [1]. To estimate the automatic engagement, all the listed components are obtained through different modalities. To improve the learning experience, researchers attempt to propose effective estimation methods for SE [4].

The widespread tools to measure SE are divided into three categories: (1) self-reports, (2) observational checklists, and (3) automated measurements. Self-reports are questionnaires that report the level of activeness, excitement, or boredom of the students during learning sessions, but they can be costly or biased tools to detect SE [5–7]. The observational checklist measures SE while relying on the questionnaires filled in by external instructors or observers. Considering some parameters, the observers rate the students' attention in live sessions or examine pre-recorded lecture videos of students but this is not a useful tool for a larger number of students [2]. Automated measurement is the recent focus of researchers that have been studying the real-time computation of SE, starting from an automated engagement tracing method that is presented for estimating SE while considering the timing and accuracy of students' responses [8]. Another automated engagement method measures students' level of arousal or alertness by using physiological and neurological sensors including EEG, blood pressure, heart rate, or galvanic skin response [8–10]. These proposals used the components of cognitive engagement for detecting SE but due to high cost and scalability issues for the real classroom environment, these methods are not useful.

Currently, researchers work only on behavioral and emotional engagement through deep learning and computer vision techniques [11]. Computer vision techniques are used to detect student engagement through their physical (apparent) or emotional behavior during lecture procession [12]. Several computer vision studies have been conducted to examine students' engagement in the e-learning (online) environment [13–15]. In such studies, behavioral indicators including the number of questions asked by the lecturer, the number of logins to the portal, the number of lectures taken, and the number of times participated in online discussions are incorporated as features to measure SE [16]. In another study, student engagement (affective state) is computed using voice recognition or physiological pressure sensors and heart rate; however, speech recognition is not recommended for affective state computation [17–19]. In the latest research, end-to-end approaches are used to detect the level of engagement of students in an online class environment through their head movement or eye gaze by using a webcam [20–22]. In subsequent work, a hybrid convolutional architecture is proposed to detect students' affective states using cues such as hand gestures, facial emotions, and body postures from an asynchronous learning environment. In this work, not only emotional but behavioral patterns are also analyzed but the trained model only provides the aggregate result of all student's states in each frame. Effective SE necessitates being computed at the individual level which may subsequently be aggregated depending upon the requirements. However, this research is conducted in a constrained environment in which students are aware of the experiment [23]. To date, there is no such method proposed that can detect SE in the offline classroom without a constrained environment, and if available, it only considered limited key features.

### 1.1. Background

Parameters such as timestamp, gender, and other demographic features may also impact the SE in the real classroom environment. It has been observed that learning during different sessions such as morning or evening may also impact SE [22]. More precisely, it is found that studying in the morning positively impacts and studying in the evening negatively impacts the student's academic performance; however, these results were reported while considering only questionnaires or surveys but not the visual features and actions of students [24–26]. Therefore, the real-time objective assessment for such findings in a classroom environment still requires the attention of the researchers. To

identify the impact of these parameters, firstly we need to detect SE in the classroom during lecture procession; then, we can perform such analysis based on engagement [23,27,28]. There are several methods for measuring active state, but all these methods have constraints as mentioned above. Detecting timestamp/gender influence on students' affective state may help improve the learning experience for students in the future. So, it is important to analyze the affective states in the real classroom environment. Furthermore, objective assessment and analysis (using visual features and action space) may also provide strong evidence as compared to subjective tools (e.g., surveys or questionnaires). Therefore, in this paper, at first we collected the data of the students during lecture processions in a classroom environment using high-definition cameras during different timestamps (morning and evening). Later, we extracted the relevant student data from the recordings into frames and further annotated it into two categories: (1) engaged and (2) non-engaged based on their emotional and behavioral patterns to train the model. Using the transfer-learning-assisted trained model, the affective states of students were computed, and subsequently, we computed the overall student affective score for engagement and non-engagement during the different sessions (morning and evening). Moreover, we also computed the student affective state score gender-wise and timestamp-wise. In the end, we compared their affective score in different timestamps and evaluated the results using inferential statistics. Thus, the primary objective is to compute the students' affective states in the real classroom environment and subsequently to analyze students' engagement gender-wise as well as in distinct timestamps. This work aims to respond to research questions including: How to compute the affective state of many students in the offline classroom environment? How can transfer learning assist to compute a model for the extraction of students' affective state? Whether male or female students remain more engaged during lecture procession? Which time of day is better for scheduling classes having more male and female students? What are the takeaways of the underlying research in terms of policy recommendations?

### 1.2. Major Contributions

Precisely, the major contributions of this research are listed as follows:

1. We collected a dataset of 45 students in a total of 32 videos from an offline and least controlled classroom setting. The extracted frames from these videos were classified into engaged and non-engaged frames based on features extracted from literature and student survey.
2. A transfer-learning-assisted model is presented to compute the affective state in an offline classroom environment while attaining surpassing correctness.
3. The explicit contribution is the subsequent analysis in which 14 different experiments are performed with respect to timestamps and six different experiments are performed to evaluate the impact of gender while incorporating Poisson and Negative Binomial Regression models.
4. The policy recommendations are suggested regarding lecture schedules of male and female students and variation in contents of the course considering findings of the underlying research.

### 1.3. Paper Organization

The remaining contents of this work are structured as follows. Section 2 provides a briefing about related literature while highlighting the gaps and presenting solutions. Materials and methods are presented in Section 3. Section 4 elaborates on the results, while the conclusion, limitations, and future suggestions are discussed in Section 5.

## 2. Literature Review

Traditionally, exams or written tests are used to assess students' academic progress. [28]. Research studies have shown that conventional methods of assessing students may negatively affect the learning process [29]. Furthermore, alternative evaluation techniques may favor learning and understanding instead of memorization [30]. Students' assessment

outcomes may also be predicted while measuring the SE in a classroom environment. Their SE level indicates how they feel while they learn, and its effective computation may assist in enhancing learning. After correctly identifying their engagement level in the lecture procession, the instructor may be able to change the way of teaching or the contents of the course for the future [31]. The engagement theorist identifies two ways through which the engagement level of a student can be determined: one way is to consider internal factors which inform analysis of the cognitive behavior, and the other is to examine external factors which include facial features, speech, actions, and postures [32]. To identify SE while employing external features, computer vision-based approaches may perform effectively, since these approaches may track facial expressions, eye movement, and body postures [33]. Even though computer-vision-based methods follow objective criteria, they have long been not practiced for the assessment process due to complex algorithms and constrained computational machines. These methods work similarly to the teacher's observation during the lecture without interrupting students' activities. To detect learner engagement using computer vision methods, visual sensors are now available in cell phones, computers, and even automobiles that can be utilized for monitoring purposes [34].

A variety of automatic engagement detection methods are presented by numerous research articles in e-learning environments that incorporate computer vision techniques [2,33]. Due to the direct relationship between facial emotions and perceived attention, SE can be determined from students' facial gestures [35]. Facial expression detection through visual sensors provides a continuous way of capturing face images during learning. Numerous automated techniques are now available to recognize and examine facial emotions and expressions [36]. An automated method for recognizing facial expressions was presented in 2018 to measure emotions and used in e-learning applications. Such SE detection methods may also assist teachers in modifying their instruction methods for students while monitoring their level of participation [37]. In a study employing a Support Vector Machine (SVM), facial emotions were recognized from video to monitor the interactivity of the lectures [38]. A method was also proposed for detecting students' emotions in the classroom using a deep convolutional neural network [39]. A teacher's automatic evaluation may also be done from video by detecting facial expressions in the classroom [40]. In 2015, in a virtual laboratory environment, the emotional rate of anger, surprise, happiness, and sadness was examined [38]. In this study, an intelligent system based on the web was proposed that helps students improve their learning process. A single face emotion detection system was proposed in 6D space, in which the teacher and student computer agent communicate with each other through an emotion detection system, and the teacher modifies their teaching style according to the student response [37]. In a similar work, video emotion recognition was proposed by combining hybrid and multi-model elements that include open EAR, LBP-TOP, and CNN features [38].

A deep learning model was introduced in another work that classifies engaged and non-engaged students [36]. This model was trained on 4627 engaged and non-engaged samples. It was the first model for engagement computation using a histogram and SVM [39]. In a similar study, the SE level was assessed by human observers, who then divided the results into four categories. This study utilized two-time timescales; one was a 10 s video and the other was a 60 s video, and the study verified (Pearson r = 0.85) that a 10 s video clip was enough to detect the students' engagement level [2]. It is argued that facial expressions are not enough to observe the student's behavioral and emotional patterns during learning. The alternative way is to recognize facial expressions along with physical facial and body actions.

In 2020, a hybrid convolutional neural network architecture was proposed that detected the affective state of the student in a classroom environment. Two models were used in the proposed architecture; one model was for detecting the SE of a single student, and the other was for detecting the SE of multiple students in a single frame. This model incorporated the facial emotions, body postures, and hand gestures of students to analyze students' affective states. Inception V3 was used to train the hybrid model. However, this

model provides the aggregated score of all the states that are detected in each frame, but the training and testing dataset was generated through the controlled environment in an offline classroom environment [23]. Later, another research work was conducted that detected the engagement level of students using ResNet, temporal convolutional network, and neural Turing machine. These tools used already available datasets such as DAiSEE and EmotiW for the training as well as for the testing. However, up to 65% accuracy is achieved due to the constrained training dataset [20–22]. Advanced machine learning techniques have been introduced such as recurrent neural networks, RCNN, Encoder-Decoders, and long short-term memory (LSTM)-based attention networks are used for relation extraction time-series classification and Hybridizing LSTM contributing to the best level in some of the latest research, such as efficient federated distillation learning system (EFDLS) [41–43]. These advanced techniques could help in the future to improve the accuracy of the current study.

**Table 1.** Comparison of affective states-related work.

| Year | Affective States | Classifier/Method | Key Features | Dataset | No. of Students | Results | Offline |
|------|------------------|-------------------|--------------|---------|-----------------|---------|---------|
| 2014 [35] | not engaged, nominally engaged, engaged, very engaged | Linear regression, multinomial logistic regression | Only head pose and eyes features | Self-generated | 34 (9 male, 25 female) | F-score: 0.369 | × |
| 2015 [44] | low, medium, and high attention levels | SVM | Head movement patterns | Self-generated | 35% female & 65% male | ACC: 0.89 | √ |
| 2017 [45] | engaged and distracted | SVM, logistic regression | Head pose and eye gaze | Self-generated | 10 (3 male, 7 female) | ACC: 90% | × |
| 2019 [12] | not engaged, normally engaged, and highly-engaged | CNN | Facial Action Unit | DAiSEE Dataset [46] | 112 (32 females and 80 males) | ACC: 89% | × |
| 2020 [23] | engaged, non-engaged, neutral | Inception v3 | Facial expressions, hand gestures, and body postures | Self-generated | 50 | ACC: 86% | × |
| 2021 [21] | low level, high-level engagement | LSTM and TCN, fully-connected neural network, SVM, and RF | Eye movement, gaze direction, and head pose | DAiSEE [46] and EmotiW [47] | 112 (32 females and 80 males) | ACC: 63% | × |
| 2021 [22] | completely disengaged, barely engaged, engaged, and highly engaged | Neural Turing Machine | Eye-gaze features, FAU, head pose, and body pose | DAiSEE [46] | 112 (32 females and 80 males) | ACC: 61% | × |

Besides the work to compute and monitor SE levels for improving learning, post-analysis is also required. For instance, the timestamp of the day in which students usually remain more focused on learning, the impact of gender on SE, etc. The timestamp at which the students feel comfortable learning depends on their chronotype. The morning chronotype person usually sleeps and wakes up early, while the evening chronotype person usually sleeps late and therefore wakes up late in the morning. Several studies have been conducted on chronotype students for different purposes [39–41]. One of the types of research was conducted on high school students' entrance tests while considering their chronotype gathered through a questionnaire. The results concluded that the morning chronotype students performed better than the evening chronotype students [48]. In another similar work, the relationship between gender and student engagement was studied but at different university levels. These studies presented mixed results and concluded that results were related but varied from university to university [49]. Several other studies were conducted to check whether the chronotype impacts academic achievements or not.

These studies concluded that it does not directly impact but rather indirectly affects learning approaches [50]. Hence, we may say that SE is impacted by chronotype, but it has not been detected through emotional and behavioral patterns in real classroom environments. The inadequate student affective state computation may lead to constrained insights. This may be misleading for the demographic analysis. Hence, it may become a hindrance to decision making regarding student learning. The effective computation of SE in a large offline classroom environment requires the attention of the research community. Moreover, the impact of timestamps and gender on students' affective states is yet to be explored. Hence, this work aims to assist decision making regarding student learning in educational institutions. In addition to these matters, some students need a particular environment to study which can also have an impact on their engagement, but different methodologies can be used to enhance their learning process [51,52], but our proposed work considers all ordinary students. Although the proposals are nuanced, the presented work overcomes several limitations discussed in the aforementioned section of objective computation of SE and subsequent gender-wise and timestamp analysis.

## 3. Materials and Methods

For computing SE and subsequently analyzing the impact of session and gender on student affective state, we recorded 32 videos in the offline classroom environment and subsequently extracted individual students' frames from the recorded videos. The extracted frames are annotated into two categories. The students in engaged states are separated as engaged frames and the students in non-engaged states are annotated as non-engaged frames. During the training, the annotated frames are trained through transfer learning using the VGG16 model by extending 4 layers. After training, testing is performed on unseen data. After modelling, the affective states of each student are computed from videos collected from the offline classroom environment. Finally, regression analysis is applied to check the impact of timestamps and gender on the students' affective states. The methodology is presented in Figure 1.

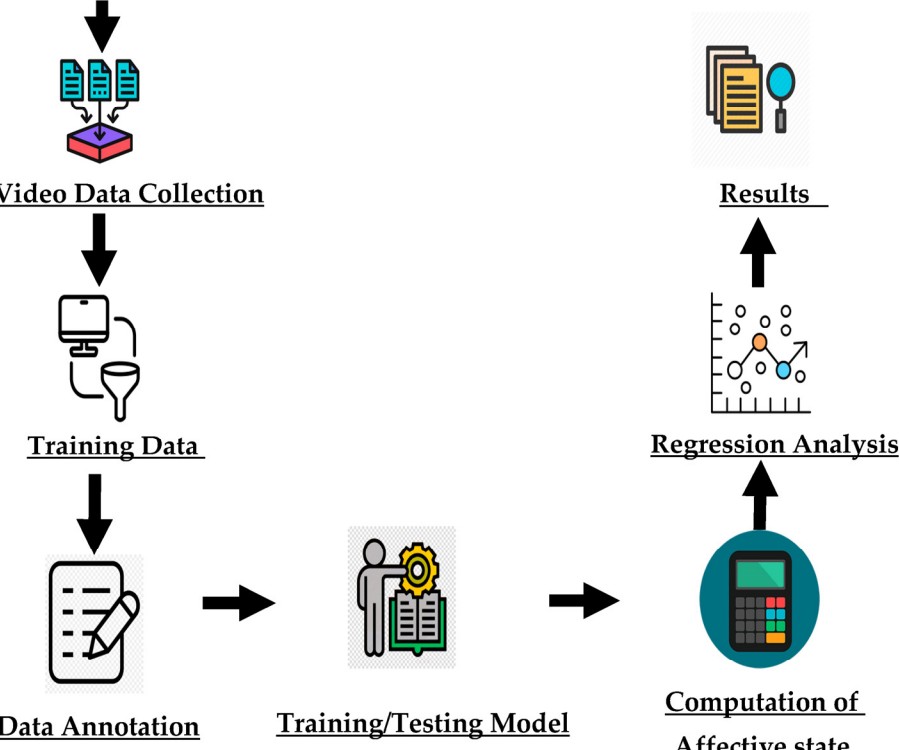

**Figure 1.** Methodological overview of the proposed system.

### 3.1. Data Acquisition

The dataset during the lecture procession in an offline classroom environment was required for both training and testing since there is no benchmark dataset available for underlying settings. For this purpose, we collected a dataset from two classes. To explore the impact of timestamps, these classes were scheduled in the morning and evening sessions. Both genders participated voluntarily to conduct the underlying research. The subject being taught was the same for both classes from the same teacher. We captured the videos of students in an offline classroom environment using two high-definition visual sensors of quality 1080p with a rate of 30 frames per second. Two visual sensors were used to monitor student engagement in two rows within a single classroom. On average, each recorded video comprised approximately 40 min. After more than one month, in the end, we collected a total of 32 videos from both classes. Due to the constraints of computational resources, two random clips from each video lecture were extracted to train and test the model. For this purpose, we processed each video and collected two random clips of 60 s from each video. In the end, we collected 64 video clips on which we performed our analysis. We divided these videos into four groups; one group contained the morning dataset and the other contained the evening dataset. Moreover, one group contained the female dataset and the other contained the male dataset. Figure 2 shows the experiment setup in the offline classroom environment.

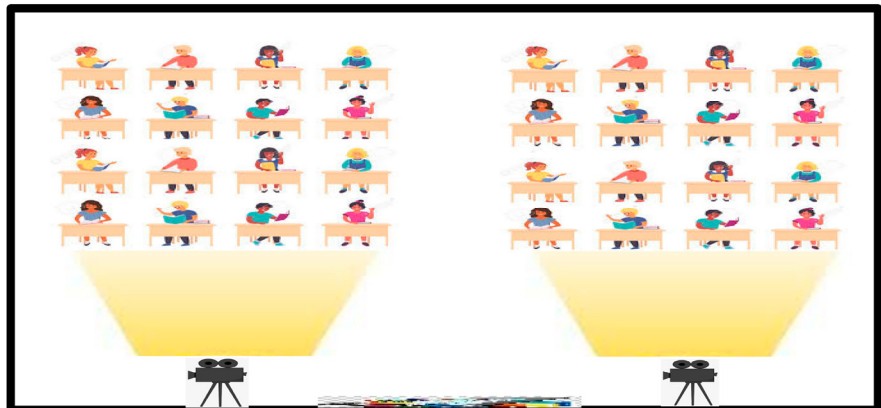

**Figure 2.** A visual diagram of experimental setup to analyze students' engagement state.

### 3.2. Model Training

To train our model, we extracted individual students' frames from the collected videos. More precisely, we used open-cv and face-recognition libraries of Python to extract the frame of each student in one second. We collected 3000 frames of all students and subsequently annotated these frames as engaged and non-engaged.

### 3.3. Data Annotations

The extracted frames for training were divided into two groups. One group was composed of engaged students and the other one included non-engaged students' frames. We annotated approximately 3000 frames after removing repeated frames for both categories for training data. To annotate the data into engaged and non-engaged categories, we took the features from the literature [24]. Moreover, we also surveyed different classes while asking the students about their gestures when they were engaged or non-engaged. We listed the most salient features to annotate the data. Figure 3 shows a sample of annotated data from engaged and non-engaged features.

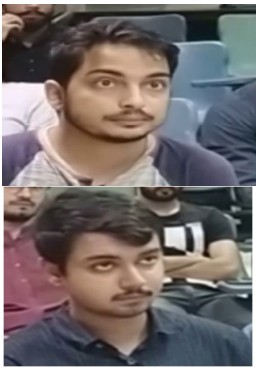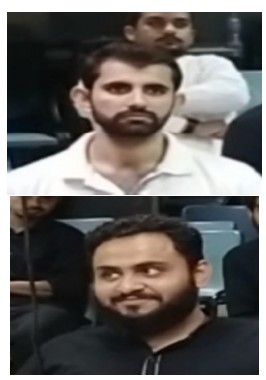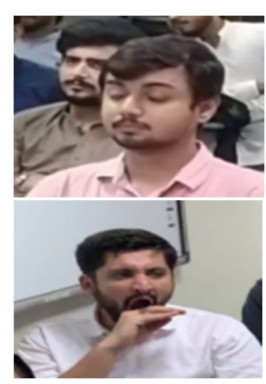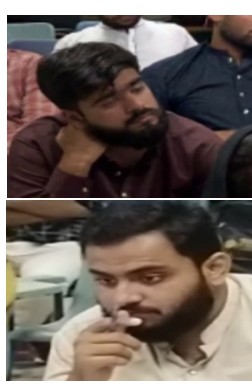

**Figure 3.** Extracted frames of engaged students are presented on the left side, and extracted frames of non-engaged students are depicted on the right side.

The incorporated features for engaged and non-engaged are briefed as follows:

1.  Engaged Frames: The frames in which the student is looking towards the teacher or board, taking notes, or discussing with a teacher are labelled as engaged.
2.  Non-Engaged Frames: The frames in which the student seems not interested in the lecture, is looking away from the teacher, barely opening or closing their eyes, yawning, leaning on the desk, using a mobile phone, or talking with fellows are labelled as non-engaged.

### 3.4. Proposed Transfer Learning Model

The objective of deep learning algorithms is to build a model for the underlying task. The details of the steps are described in Algorithm 1. However, building and training a model from scratch is not an easy task, since many hyperparameters are to be tuned. Moreover, a large dataset is required for building the model. In such cases, transfer learning may help, where we can employ the learning of a standard model trained on some standard dataset for some relevant problem. Deep learning models extract relevant features from the given data and attain state-of-the-art correctness that sometimes even surpasses human performance [53]. In our research, we incorporated a variant of Convolutional Neural Network (CNN) to build our model for the underlying task. The CNN variants are the most popular DL architectures, and their popularity is increasing day by day due to their practical effectiveness [54]. CNNs have many pre-trained learning models including Inception, ResNet and VGG [55], and others [56]. We trained inceptionv3 and VGG16 models on our collected dataset but VGG16 gives back better results by adding extended dense layers on the tested dataset. Therefore, we incorporated the VGG16 model for classifying whether a particular student in the video is engaged or non-engaged. Originally, the VGG16 was trained for object detection and classification from the image. It is a multi-classification network, but we used this model for binary classification. VGG16 employs 16 layers. The three fully connected layers follow a stack of convolutional layers that have different depths in different architectures and the final layer is the classification layer. First, convolutional layers are designed to extract a maximum number of features from the given image. Images are usually fed to this network with a dimension of (224 × 224) pixels.

In fact, we did not use specific hand-crafted features in this article as presented in Table 1. We used a straightforward VGG16 model with extended and dense layers to extract maximum visual features from human faces. Those features are already trained by using a pretrained VGG16 model based on engaged and non-engaged classes. Compared to state-of-the-art studies, it is very difficult to extract meaningful and manual features for classification tasks.

**Extended Layers:** To build our model, we used the settings of VGG16 and trained the last two layers of the pre-trained model. Additionally, four more layers were added for modifying the VGG16 model for our classification task. The settings of the four new layers include a Flatten layer, a Dense (128) layer, a Dense (64) layer, and a Dense (1) layer. Flatten

layer was added to get the results from the pre-trained layers of the model for further processing. Dense (128) represents a hidden layer with 128 neurons, Dense (64) represents another hidden layer with 64 neurons and the final layer is Dense (1) which represents the output layer. The training status of the remaining layers was set to False.

**Activation Function:** To scale out the results, we used the 'sigmoid' activation function as sigmoid performs better in binary classification problems. Figure 4 shows the overview of the modified VGG16 architecture.

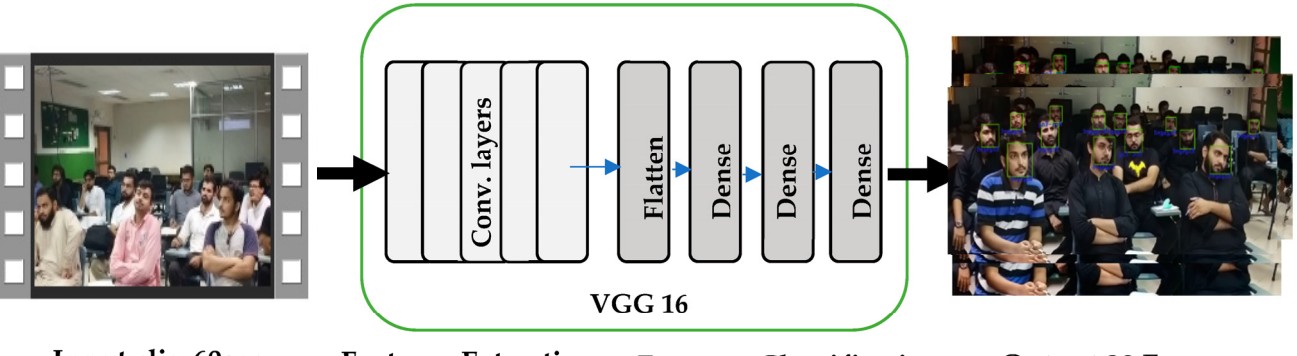

**Figure 4.** Overview of the designed proposed VGG16 with extended layers architecture to recognize engaged and non-engaged students.

| Algorithm 1: That proposed VGG16 with dense layers and fine-tuning of the model and hyperparameters |
| --- |
| Step 1: *Input: Video frames with annotations (engaged or non-engaged) and—Timestamps and gender information for each student.* <br> Step 2: *Output: Affective states of each student (engaged or non-engaged) based on the video frames and Regression analysis results (impact of timestamps and gender on affective states).* <br> Step 3: *Data Collection: Let 'X' be the set of video frames with annotations and Let 'Y' be the corresponding labels for each frame (1 for engaged, 0 for non-engaged).* <br> Step 4: *Data Preprocessing: Resize and preprocess the frames in 'X' to a standardized size and split the data into training and testing sets: 'X_train', 'Y_train', 'X_test', 'Y_test'.* <br> Step 5: *Transfer Learning with VGG16 and Fine-tuning:* <br>     (a) *Load the pre-trained VGG16 model with weights learned on a large image dataset (StudentEng-NonEng).* <br>     (b) *Add or replace the final layers to match the number of classes (engaged and non-engaged).* <br>     (c) *Add dense layers* <br>     (d) *Optionally, unfreeze some of the later layers to fine-tune the model on the new task.* <br> Step 6: *Repeat for layer in VGG16.layers[:-4]: Layer.trainable = False* <br> Step 7: *Define the loss function: Loss(Y_true, Y_pred), e.g., categorical cross-entropy, and define the optimization algorithm: Optimization Algorithm with appropriate hyperparameters (e.g., learning rate, momentum).* <br> Step 8: *Compile the model: Train the model on the training data with a batch size and number of epochs, Evaluate the fine-tuned model on the testing data, Apply the fine-tuned model to predict the affective states of each student.* <br> Step 9: *Combine affective state predictions with metadata (timestamps and gender) for each student.* |

## 3.5. Fine-Tuned Model and Hyperparameters

Transfer learning is the process of applying the acquired knowledge of a previously trained model to a new problem. Transfer learning is a popular technique in deep learning because the model has been previously trained on a large dataset, necessitating significantly fewer computational resources. Typically, the technique is implemented by transferring the learned characteristics of pre-trained models to tasks that follow. Fine-tuning is required to ensure that the pre-trained model effectively adapts to the subsequent duties. During the process of fine-tuning, the entire model or a portion of the model is defrosted. The number of dense layers and classifier layers to be added to the network depends on the difficulty of the tasks that will be performed subsequently.

In this study, a previously trained VGG-16 model [55] was refined. The VGG-16 model is an enhanced variant of the AlexNet [3] model, consisting of more convolutional layers and employing the tiniest kernel size. The VGG-16 model was trained with 15 million images from the ImageNet dataset. The VGG-16 model consists of thirteen convolutional

layers, five max-pooling layers, and three fully connected layers, with the convolutional and max-pooling layers segmented into five sets. Following two convolutional layers in the first two sets is a max-pooling layer. The next three sets are composed of three convolutional layers and a max-pooling layer. The network-wide use of a kernel size of $3 \times 3$ with stride 1 and buffering 1 distinguishes this model from other pre-trained models. Utilizing the tiniest kernel size drastically reduces the quantity of parameters. In addition, the use of the minimum kernel size prevented the network from becoming overfitted. Max pooling was conducted on a $2 \times 2$-pixel window with a stride of 2. By doing so, the original size of the feature maps is reduced by half. All convolutional layers were activated with the rectified linear unit (ReLU) activation function because it is computationally economical and reduces the vanishing gradient problem. The pre-trained VGG-16 model was selected because it is simple to implement. Additionally, fewer parameters are involved in the model, resulting in a faster-learning network.

Tuning of hyperparameters is essential for optimizing the efficacy of TL learning models. In this study, the hyperparameters were optimized by employing a grid search technique on the selected dataset. Four hyperparameters are involved in the refining process: batch size, dropout value, learning rate, and optimizer. The grid search is conducted by varying the value of a single hyperparameter at a time, while the values of the remaining three hyperparameters remain constant. Table 2 displays the evaluated and optimal hyperparameter values for the proposed method. The optimal value for each hyperparameter is determined by maximizing accuracy while minimizing computation time. Hyperparameters batch size of tested values (16, 32, 64, 128), optimized value (16), dropout Value tested values (0.2, 0.3, 0.40) optimized value (0.3), learning Rate test values (0.0001, 0.001, 0.01) optimize value (0.001), and Optimizer tested techniques (SGD, Adam) selected Adam.

**Table 2.** Optimal hyperparameters for the proposed VGG16 model.

| Hyperparameter | Optimal Value |
| --- | --- |
| Learning Rate | 0.001 |
| Batch Size | 16 |
| Epochs | 50 |
| Weight Decay | 0.0005 |
| Dropout Rate | 0.3 |
| Activation Function | ReLU |
| Optimizer | Adam |

## 4. Results and Discussions

### 4.1. Environmental Setup

For a compilation of the VGG16 model, we used 'binary_crossentropy' as a loss function because of the binary classification. We used Stochastic Gradient Descent as an optimization function with a learning rate of 0.001 and accuracy is used as the correctness measure. In the first step of training, we trained the model on different batch sizes (16, 32) and epochs (50, 100, 150, 200). However, during validation, we analyzed that training on the 16-batch size and 100 epochs performed better than the other settings. The dataset comprised 3000 images in total out of which 20% of data were used for testing and the remaining 80% were used for training and validation. In the literature, we observed that the performance of models was mostly tested through accuracy [24]. We used correctness measures including Accuracy, Precision, Recall, and F-Measure for the testing of our model. Our proposed model predicts the engaged and non-engaged frames with 90% accuracy and classified 93% of frames into positive class (recall); 93% of frames were correctly classified into both positive and negative classes (precision). Table 3 presents some important correctness measures for the computation of the affective states of students in a real classroom environment.

**Table 3.** Time and accuracy obtained for the proposed method.

| Training Platform | Training Time (h) | Testing Accuracy (%) | Testing Time |
|---|---|---|---|
| Google-Colab-Pro (16 GB GPU, 25 GB memory, 147 GB storage) | 1 | Accuracy 0.90 Precision 0.93 Recall 0.93 F-measure 0.93 | 6 s/frame |

## 4.2. Computations of Engagement State

We input each clip from the recorded lectures and applied the trained model to it. We took one frame from every two subsequent seconds to avoid the repetition of the frames since repetitive frames output the same value of the state. We applied the bounding box on the faces in each frame of students through the face recognition library. Thus, as an output, we obtained 60 computed affective states against each student. We analyzed each frame and then collected the data of engaged and non-engaged students with unique student IDs. We applied the same process to all the recorded videos and in the end, we collected 283 observations of 45 students including males and females. The total analysis was performed on 13,510 individual frames for both engaged and non-engaged students. We compare the accuracy of our proposed model with the existing methods in Table 4.

**Table 4.** Comparison of accuracy with existing methods.

| Year | Classifier/Method | Affective States | Accuracy | Offline Classroom Environment |
|---|---|---|---|---|
| 2014 [35] | Linear regression, multinomial logistic regression | not engaged, nominally engaged, engaged, very engaged | Not Reported | × |
| 2015 [44] | SVM | low, medium, and high attention levels | 62% | √ |
| 2017 [45] | SVM, logistic regression | engaged and distracted | 90% | × |
| 2019 [12] | CNN | not engaged, normally engaged, and highly engaged | 89% | × |
| 2020 [23] | Inception v3 | engaged, non-engaged, neutral | 86% | × |
| 2021 [21] | LSTM and TCN, fully connected neural network, SVM, and RF | low-level, high-level engagement | 63% | × |
| 2021 [22] | Neural Turing Machine | completely disengaged, barely engaged, engaged, and highly engaged | 61% | × |
| Proposed method | VGG16 (Extended layers) | engaged, non-engaged | 90% | √ |

## 4.3. Methods for Post Analysis

After computing SE, data were analyzed using inferential statistical methods including Poisson Regression (PR) and Negative Binomial Regression (NBR). Regression modelling techniques were used to analyze the association between multiple variables with count outcome data. Linear regression is not suitable for counting data. PR applies to the count or rate data. The count data are quantified with a count variable that is taken from discrete non-negative number values in a fixed interval. As a Cobb-Douglass production

function [55], the output is a function of inputs. In our case, we consider engagement and non-engagement as a function of session and gender by this production function.

$$\text{Eng} = f(\text{Session}, \text{Gender}) \tag{1}$$

In the generalized linear models, the response variable is a binary variable such as in the form of yes or no, 0 or 1, or A or B. Hence, the relationship between the dependent and independent variables may not be linear.

$$y_i = \alpha + \beta_i X + e_i \quad i = 1, 2, \ldots n \tag{2}$$

In Equation (2), y is the dependent variable, $\alpha$ is the constant other than factors that affect the dependent variable, $\beta$ is the coefficient, and X is the independent variable. For the analysis of our problem, we formulated the following four equations: Equations (3)–(6). More precisely, Equations (3) and (4) are modelled for an engaged variable concerning the independent variables of session and gender, respectively. Equations (5) and (6) are formulated for the non-engaged variable concerning the independent variable session and gender, respectively.

$$\text{Eng} = \text{Const.} + \beta_1 \text{Session} + \varepsilon_t \tag{3}$$

$$\text{Eng} = \text{Const.} + \beta_1 \text{Gender} + \varepsilon_t \tag{4}$$

$$\text{Non\_Eng} = \text{Const.} + \beta_2 \text{Session} + \varepsilon_t \tag{5}$$

$$\text{Non\_Eng} = \text{Const.} + \beta_2 \text{Gender} + \varepsilon_t \tag{6}$$

Although PR modelling is widely used for the analysis of count data, it does not handle over-dispersed data. PR assumes that the outcome variable follows Poisson distribution, which means that the mean and variance are equal. However, over-dispersed data implies that the mean and variance are not equal. This condition is handled through NBR because it also considers the over-dispersion of data.

### 4.4. Results Analysis

The results are obtained after applying the learned model to the recorded lecture clips. The students not interested in lectures displayed some dominant features such as using a mobile phone, sleeping, laughing with other fellows, closing their eyes, and yawning, as shown in Figure 5. Attentive students taking great interest in the lecture are detected using the employed features since such students are found to be writing notes, communicating with the teacher, and looking at the board. Figure 6 depicts engaged examples detected through the proposed model. Figure 7 depicts the results of the trained model on an unseen frame during the lecture procession. Model results depict almost all the clear faces and label them with an accurate engaged or non-engaged tag according to their detected features.

The collected data for 8 days were organized gender and timestamp-wise. We refer to males with 1 and females with 0. Similarly, we denote the morning session with 1 and the evening with 0. A total of 60 frames were collected from each clip and a total of 60 values against each student were categorized as either engaged or non-engaged. In some frames, some students were not detected because their faces were partially or completely hidden behind other students. In some cases, the students were absent, so we also ignored those values. The following graphs in Figure 8 show that the engaged data are positively skewed, and the non-engaged data are negatively skewed. More precisely, the *x*-axis shows the 60 bins representing 60 frames from each video, while the *y*-axis represents the engaged and non-engaged frames distribution of students. For instance, it may be observed from the left distribution that the highest number of engaged counts is around 35, which means that it appears 47 times that students remain engaged in 35 extracted frames out of a total of

60 frames. Similarly, the right distribution depicts that the highest number of non-engaged count is around 10, meaning that it appears 38 times that students remain non-engaged in 10 frames out of a total of 60 frames.

**Figure 5.** Accuracy versus loss with respect to train and test splits for VGG16 (**a**,**b**) model and proposed VGG16 with dense layers (**c**,**d**) model to recognize engaged non-engaged frames from testing video.

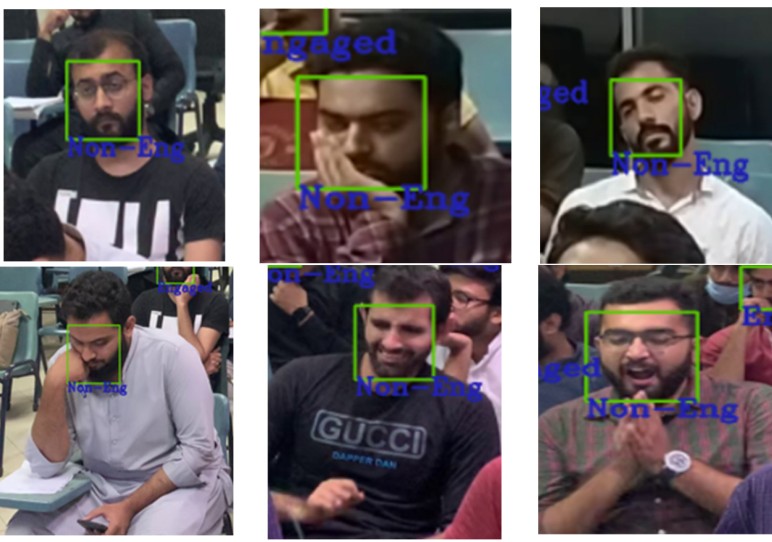

**Figure 6.** Detected non-engaged frames from testing video.

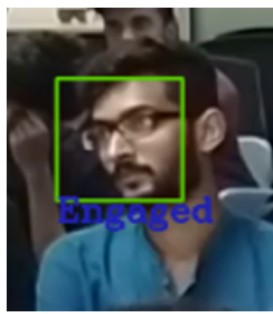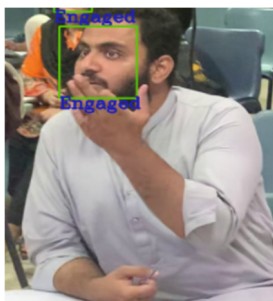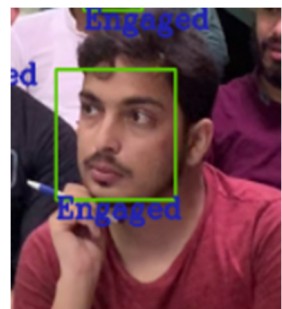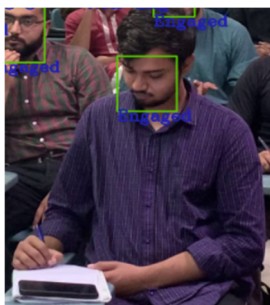

**Figure 7.** Detected engaged frames from testing video.

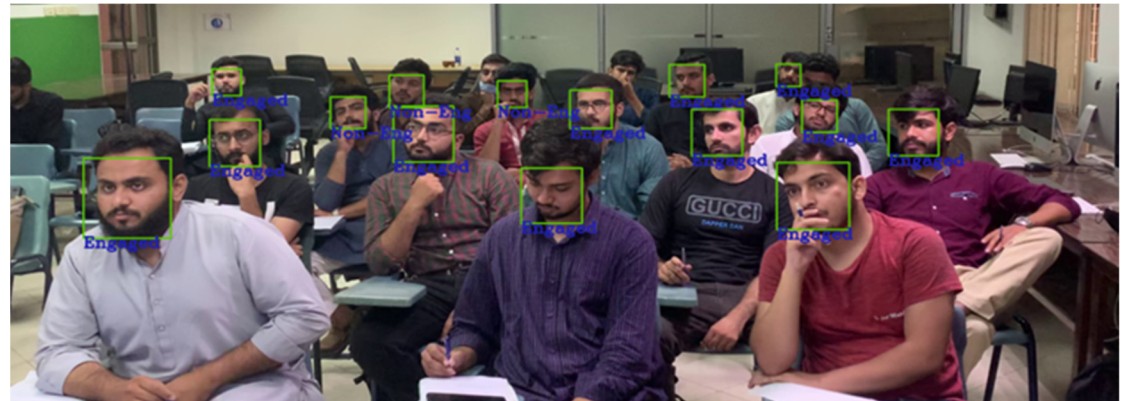

**Figure 8.** Single frame from testing video.

### 4.5. Timestep Analysis

The session is the independent variable that is analyzed through both PR and NBR to test whether there is any impact of the timestamp on SE or not. We performed multiple experiments regarding timestamps to analyze the impact of a session on SE or lack thereof. The results of the experiments are listed in Table 5.

**Table 5.** List of all experiments with respect to timestamp.

| | Experiments | Independent Variable | Dependent Variable | Results |
|---|---|---|---|---|
| 1 | Class A: MorningMale (1) vs. EveningMale (0) students with engagement | Session | Engagement | No significant impact |
| 2 | Class A: MorningMale (1) vs. EveningMale (0) students with non-engagement | Session | Non-Engagement | Male students decrease non-engagement in the morning session |
| 3 | Class B: MorningMale (1) EveningMale (0) students with engagement | Session | Engagement | Male students increase engagement in the morning session |
| 4 | Class B: MorningMale (1) EveningMale (0) students with non-engagement | Session | Non-Engagement | Male students decrease non-engagement in the morning session |
| 5 | Class A: MorningFemale (1) vs. EveningFemale (0) students with engagement | Session | Engagement | No significant impact |
| 6 | Class A: MorningFemale (1) vs. EveningFemale (0) students with non-engagement | Session | Non-Engagement | No significant impact |

**Table 5.** *Cont.*

| | Experiments | Independent Variable | Dependent Variable | Results |
|---|---|---|---|---|
| 7 | Class B: MorningFemale (1) vs. EveningFemale (0) students with engagement | Session | Engagement | No significant impact |
| 8 | Class B: MorningFemale (1) vs. EveningFemale (0) students with non-engagement | Session | Non-Engagement | No significant impact |
| 9 | All MorningMale (1) vs. EveningMale (0) students with engagement | Session | Engagement | Male students increase engagement in the morning session |
| 10 | All MorningMale (1) vs. EveningMale (0) students with non-engagement | Session | Non-Engagement | Male students decrease non-engagement in the morning session |
| 11 | All MorningFemale (1) vs. EveningFemale (0) students with engagement | Session | Engagement | No significant impact |
| 12 | All MorningFemale (1) vs. EveningFemale (0) students with non-engagement | Session | Non-Engagement | No significant impact |
| 13 | All Morning (1) vs. all Evening (0) students with engagement | Session | Engagement | Engagement increases in the morning |
| 14 | All Morning(1) vs. all Evening(0) students with non-engagement | Session | Non-Engagement | Non-engagement decreases in the morning |

The details of experiments that had a significant impact on engagement or non-engagement are explained below.

Case 1: The first experiment is performed in which all the students of the morning session are compared with all students of the evening session with respect to engagement.

**Null Hypothesis:** *The session does not have any impact on engagement.*

In this analysis, we took session as an independent variable and engagement as the dependent variable. By analyzing the Z-value (2.42), we concluded that there was a positive impact of the session on the engagement of students with 95% acceptance. Thus, the null hypothesis was rejected because it fell in the acceptance region. So, there is a significant impact such that morning students increase their engagement as compared to evening students, as shown in Table 6. NBR also verified the results of PR. Furthermore, the diagnostic test also confirms that our model fits the collected data.

$$\text{Eng} = 3.447 + 0.1011\ \text{Session} \tag{7}$$

**Table 6.** All morning (1) vs. all evening (0) students with engagement.

| Independent Variable | Dependent Variable | Const. | Coefficient | Std. Err | Z Value | Prob | Test |
|---|---|---|---|---|---|---|---|
| Session | Non-engaged | 2.782804 | −0.1842965 | 0.0639207 | −2.88 | 0.004 | PR |
| | | 2.782804 | −0.1842965 | 0.0677216 | −2.72 | 0.007 | NBR |

Case 2: The second experiment is performed in which all the students of the morning session are compared with all students of the evening session with respect to non-engagement.

**Null Hypothesis:** *The session does not have any impact on non-engagement.*



We took the session as an independent variable and non-engagement as the dependent variable. By analyzing the Z-value (−2.88), we concluded that there was a negative impact of a session on the engagement of students with 99% acceptance. Thus, the null hypothesis was rejected because it fell in the acceptance region. There is a significant impact such that morning students have a decrement in non-engagement as compared to evening students. Furthermore, the diagnostic test also confirms that our model fits the collected data, as shown in Table 7.

**Table 7.** All morning (1) vs. all evening (0) students with non-engagement.

|   | Experiments | Independent Variable | Dependent Variable | Results |
|---|---|---|---|---|
| 1 | All MorningMale (1) vs. all MorningFemale (0) students with engagement | Gender | Engagement | No significant impact |
| 2 | All Morning(1) vs. all Evening(0) students with non-engagement | Gender | Non-Engagement | No significant impact |
| 3 | All EveningMale (1) vs. all EveningFemale (0) students with engagement | Gender | Engagement | Male students decrease engagement as compared to female |
| 4 | All EveningMale (1) vs. all EveningFemale (0) students with non-engagement | Gender | Non-Engagement | No significant impact |
| 5 | All Males (1) vs. All Females (0) with engagement | Gender | Engagement | Male students decrease engagement as compared to female |
| 6 | All Males (1) vs. All Females (0) with non-engagement | Gender | Non-Engagement | No significant impact |

### 4.6. Gender-Wise Analysis

In this section, gender is taken as the independent variable to analyze against SE using both PR and NBR models to check whether there is an impact of gender on SE or not. Multiple experiments were performed regarding timestamps to analyze the impact between gender and engagement/non-engagement. The results of the experiments are listed in Table 8.

**Table 8.** List of experiments with respect to gender.

| Independent Variable | Dependent Variable | Const. | Coefficient | Std. Err | Z Value | Prob | Test |
|---|---|---|---|---|---|---|---|
| Session | Engaged | 3.447717 | 0.1011712 | 0.0417288 | 2.42 | 0.015 | PR |
|  |  | 3.447717 | 0.1011712 | 0.0473721 | 2.14 | 0.033 | NBR |

The details of experiments that have a significant impact on engagement or non-engagement are explained below.

Case 1: All male students are compared with all the female students with respect to engagement.

**Null Hypothesis:** *Gender does not impact engagement.*

For this analysis, gender is taken as an independent variable, and engagement is the dependent variable. It may be observed that the Z-value of both the PR and NBR lies under the acceptance region and rejected the null hypothesis as shown in Table 9. Therefore, it is

concluded that there is an impact of gender on engagement but the negative relationship means that male students decrease their engagement as compared to female students.

$$\text{Eng} = 3.568 - 0.0985 \text{ Gender} \tag{8}$$

**Table 9.** All males (1) vs. All females (0) with engagement.

| Model | Epochs | ACC |
|---|---|---|
| VGG16 | 40 | 79 |
| AlexNet | 40 | 81.3 |
| InceptionV3 | 40 | 82.7 |
| GoogleNet | 40 | 83.5 |
| Xception | 40 | 82.4 |
| MobileNet | 40 | 84.3 |
| SqueezeNet | 40 | 87.6 |
| Proposed VGG-16 | 40 | 90.01 |

Case 2: All male students are also compared with all the female students with respect to non-engagement.

**Null Hypothesis:** *Gender does not impact non-engagement.*

In this case, gender is taken as an independent variable, and non-engagement is the dependent variable. The results reveal that there is an insignificant impact of gender on non-engagement because the z-value does lie in the acceptance region as shown in Table 10. Therefore, the null hypothesis is accepted.

**Table 10.** All males (1) vs. all Females (0) with non-engagement.

| Independent Variable | Dependent Variable | Const. | Coefficient | Std. Err | Z Value | Prob | Test |
|---|---|---|---|---|---|---|---|
| Session | engaged | 3.568845 | −0.0985885 | 0.0417993 | −2.36 | 0.018 | PR |
| | | 3.568845 | −0.0985885 | 0.0520437 | −1.89 | 0.058 | NBR |

The generalizability of the proposed model is validated using an independent experiment. For this purpose, some clips of male and female students in the offline classroom environment were recorded and tested using the proposed model. These students were not incorporated for training or even for the testing phase of model building. Figure 9 presents one frame from the testing video through which we may observe that all students are detected, and the proposed model labeled them according to their engagement and non-engagement states. The attained average accuracy for this independent experiment on unseen data is found to be 83%, which is close to the 90% of model accuracy, so the absolute error is 7% which can be ignored. Hence, the generalizability of the proposed model is validated. Our proposed model can be embedded with a hardware system to be deployed in a real-time classroom environment to detect the engagement and non-engagement state of the students.

We compared various TL algorithms and models, including VGG16, AlexNet, InceptionV3, GoogleNet, Xception, MobileNet, SqueezeNet, and the proposed model. The classification results of the pretrained DL models for batch size 16 was calculated and presented in Table 10. The efficacy of the developed VGG16 extended with fine-tuned model achieved reasonable accuracy. However, group size had an influence on the number of parameters and computational time. However, the classification results for other pre-trained TL algorithms remain unchanged.

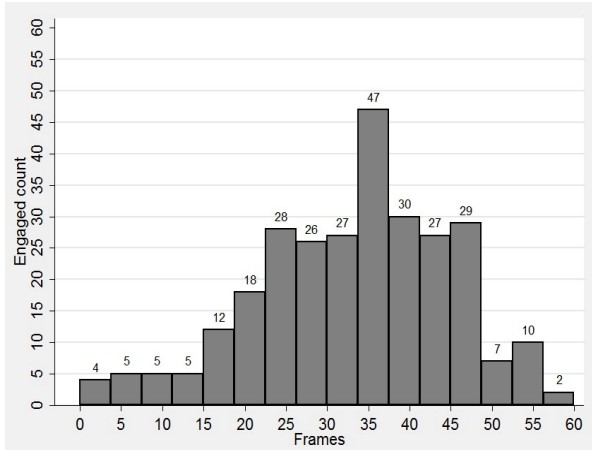 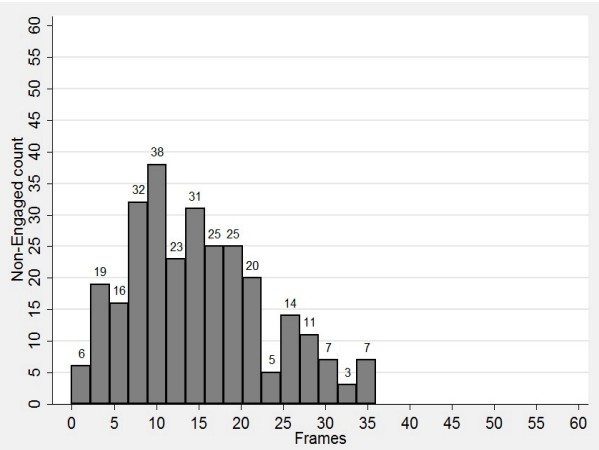

**Figure 9.** Engaged and non-engaged distribution.

In addition, Table 11 and Figure 10 demonstrate that the extended model of VGG16 is outperformed compared to other TL models in detecting multiple objects from live video camera scenes. Here are a few reasons why VGG16 may be outperformed by these architectures for object detection in videos:

1.  Spatial information vs. temporal information: VGG16 focuses on capturing spatial information within individual frames, but it does not explicitly model temporal dependencies between frames. In contrast, architectures such as InceptionV3 and GoogLeNet incorporate components such as temporal convolutional layers or recurrent neural networks (RNNs) that can capture temporal information and dependencies in video sequences. This can be beneficial for object detection in videos, where the motion and temporal context of objects play an important role.

2.  Computational efficiency: VGG16 has a relatively high number of parameters and computational complexity due to its deeper architecture, which can make it computationally expensive for real-time object detection in videos. InceptionV3 and GoogLeNet, on the other hand, have been designed with computational efficiency in mind. They utilize techniques like $1 \times 1$ convolutions and factorized convolutions, which reduce the number of parameters and computational cost while maintaining or even improving performance. This efficiency is particularly advantageous for video processing tasks that require real-time or near-real-time performance.

3.  Architectural innovations: InceptionV3 and GoogLeNet incorporate architectural innovations that aim to address specific challenges in object detection, such as the problem of vanishing/exploding gradients or the efficient use of network capacity. These innovations, such as the use of inception modules, auxiliary classifiers, and reduction layers, can enhance the model's ability to detect objects accurately in videos.

**Table 11.** Results of the proposed system model's classification using 16 batches of data.

| Independent Variable | Dependent Variable | Const. | Coefficient | Std. Err | Z Value | Prob | Test |
| :---: | :---: | :---: | :---: | :---: | :---: | :---: | :---: |
| Gender | Non-engaged | 2.720974 | −0.0379002 | 0.0682888 | −0.55 | 0.579 | PR |
| | | 2.720974 | −0.0379002 | 0.0753602 | −0.50 | 0.615 | NBR |

*4.7. Generalizability Analysis*

The generalizability of the proposed model is validated using an independent experiment. For this purpose, some clips of male and female students in the offline classroom environment were recorded and tested using the proposed model. These students were not incorporated for training or even for the testing phase of model building. Figure 9 presents one frame from the testing video through which we may observe that all students

are detected, and the proposed model labeled them according to their engagement and non-engagement states. The attained average accuracy for this independent experiment on unseen data is found to be 83%, which is close to 90% of model accuracy, so the absolute error is 7%, which can be ignored. Hence, the generalizability of the proposed model is validated. Our proposed model can be embedded with a hardware system to be deployed in a real-time classroom environment to detect the engagement and non-engagement state of the students.

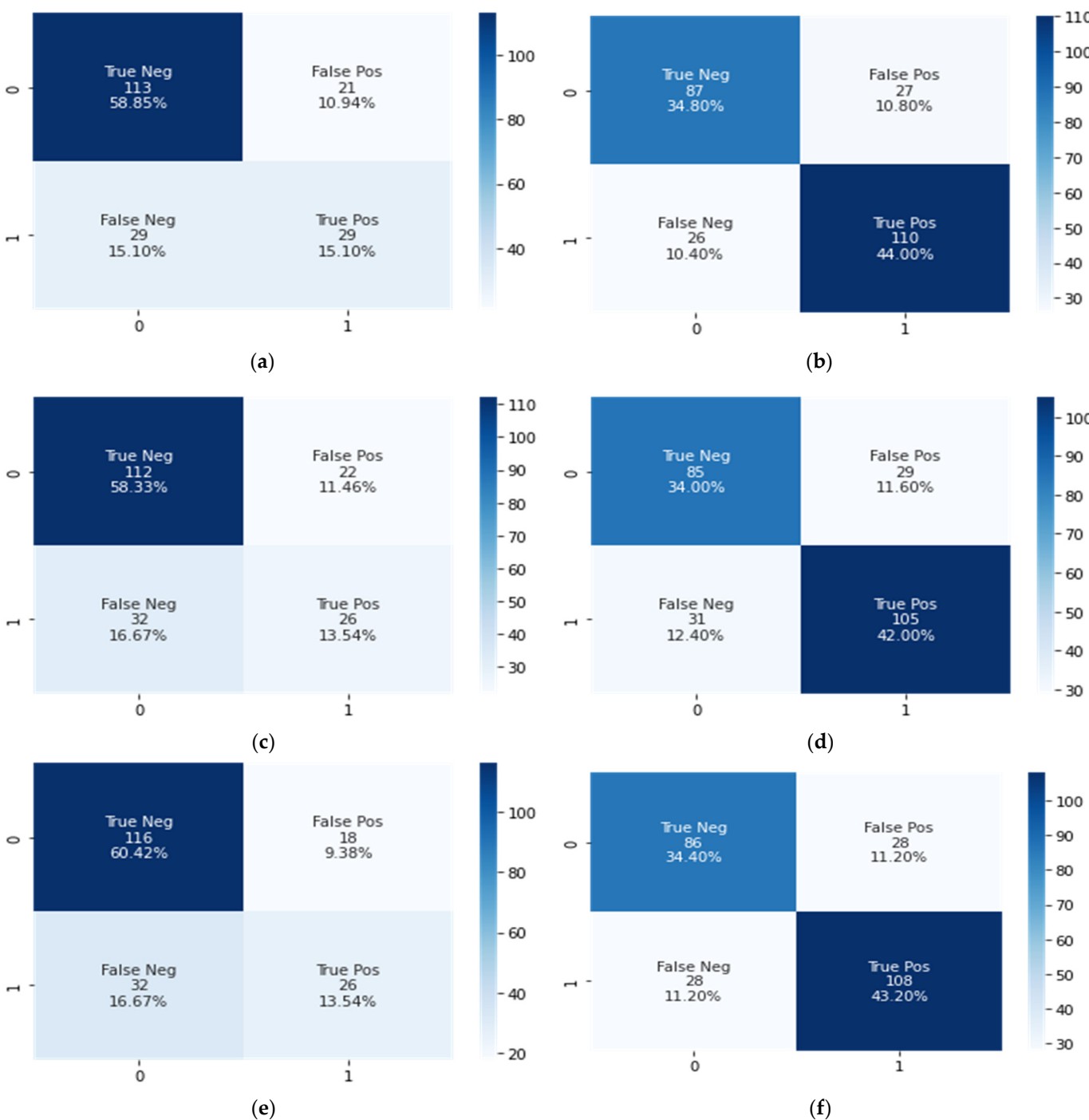

**Figure 10.** Confusion matrix represents results obtained from Engaged and non-engaged distribution with 0-index and 1-index, respectively, where figures show (**a**) Neural Networks, (**b**) CNN, (**c**) LSTM, (**d**) InceptionV3, (**e**) VGG16, and (**f**) proposed VGG16-Dense model.

A comparison in Table 12 shows the key characteristics of various machine learning algorithms such as SVM (Support Vector Machines), Random Forest, Neural Networks, CNN (Convolutional Neural Networks), LSTM (Long Short-Term Memory), InceptionV3, and VGG16 for object recognition in video scene analysis. This table shows the advantages of the proposed VGG16 model with extended layers compared to state-of-the-art machine learning algorithms. A random video clip is also visually represented in Figure 11 to show the detect human faces with engage and non-engage states of students.

**Table 12.** A comparison table outlining the key characteristics of various machine learning algorithms such as SVM (Support Vector Machines), Random Forest, Neural Networks, CNN (Convolutional Neural Networks), LSTM (Long Short-Term Memory), InceptionV3, and VGG16.

| Model | Disadvantages |
| --- | --- |
| SVM | -Limited ability to capture complex relationships in data |
| | -Requires feature engineering |
| | -May struggle with large-scale datasets |
| Random Forest | -Can be computationally expensive |
| | -May require tuning of hyperparameters |
| | -Prone to overfitting with noisy or imbalanced datasets |
| Neural Networks | -Requires large amounts of labeled training data |
| | -Computationally intensive, especially for deep architectures |
| | -Prone to overfitting without proper regularization |
| CNN (Convolutional Neural Networks) | -Requires large amounts of labeled training data |
| | -Computationally intensive, especially for deep architectures |
| | -Prone to overfitting without proper regularization |
| LSTM | -Requires longer training times |
| | -Can be more complex to implement compared to other models |
| | -Prone to vanishing/exploding gradient problems |
| InceptionV3 | -May not perform as well with limited training data |
| | -Can be computationally expensive for real-time applications |
| | -Limited ability to model long-term temporal dependencies |
| **Compare to original VGG16, we have provided the following benefits of the proproposed VGG16-dense architecture as follows:** | |
| VGG16-Dense | -Deep architecture for capturing intricate image features |
| | -Transfer learning capabilities |
| | -Suitable for image-based tasks |

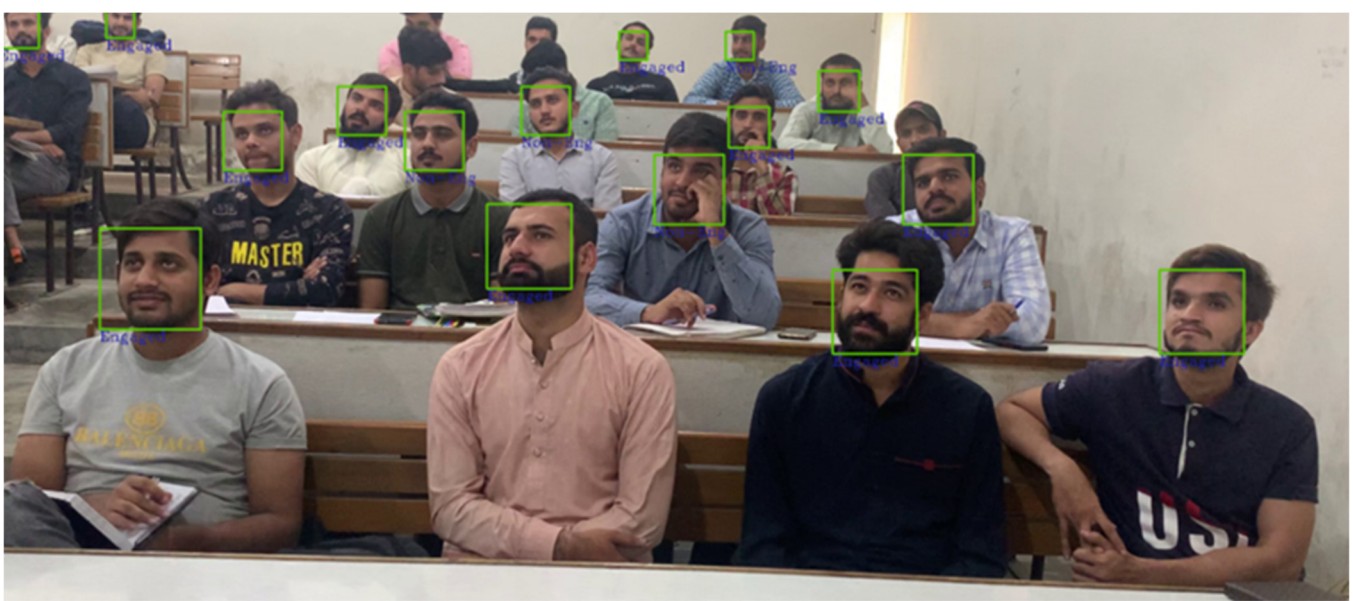

**Figure 11.** Tested frame from a random clip.

*4.8. Computational Analysis*

To determine the computational complexity of our proposed VGG16 deep learning model, we need to consider the number of operations required to process each frame. Here is an estimation of the computational complexity based on the information provided:

1.  Frame Down sampling: We process one frame after two seconds.
2.  Image Size: Each image is converted to a size of $224 \times 224$ pixels before being passed to the model. This means each frame consists of $224 \times 224 \times 0.5$ (RGB channels) = 25,088 input values.
3.  Model Inference: The computational complexity of these layers can be estimated based on the number of operations required for each layer type. However, the exact number of operations can vary depending on the specific architecture and implementation details.

As a rough estimate, let us consider a simple CNN architecture with a few convolutional layers and fully connected layers. Assuming a total of N operations are required for the model inference per input frame, the computational complexity for each second of the camera stream (0.5 frames) would be approximately 0.5N operations.

## 5. Conclusions and Future Works

This bifold research presented a transfer-learning-assisted model for measuring students' affective states. For this purpose, we trained the model using prescribed engaged and non-engaged features of the students. The overall average accuracy of the model is found to be 90%. We also performed an independent experiment for proving the generalizability of the proposed model on unseen video and achieved 83% accuracy. Finally, the inferential statistic was employed to check the impact of both timestamps and gender on students' engagement. The findings show that gender and timestamp have a substantial impact on students' participation in an offline classroom environment. In contrast to the evening sessions, the morning sessions show higher levels of student engagement. According to the gender analysis, females are more likely to stay engaged than males. The research findings informed the following recommendations:

*   A class having more male students is better to be scheduled in the morning.
*   A class having more female students may also be scheduled in the evening.

*5.1. Limitations*

The impact of timestamps and gender is analyzed on a relatively small dataset. The presented model can take only one video at a time to compute students' engagement. We only considered their facial actions and hand gestures for the underlying experiment.

*5.2. Future Work*

The dataset may be enhanced while considering more students in an offline classroom environment. To analyze the affective states, more features may be added to categorize them into engaged and non-engaged states; in addition, other states such as neutral states may also be considered. Facial emotions and body postures may also make a difference. Audio speech cannot be detected through a webcam although we can use visual speech to detect the engagement of students. Moreover, more transfer learning models and self-built architectures may be explored for the computation of effective states.

**Author Contributions:** Conceptualization, H.A., N.M., C.M.N.F., Q.A., I.Q. and A.H.; Data curation, H.A., N.M., C.M.N.F., I.Q. and A.H.; Formal analysis, S.I., H.A., I.Q. and A.H.; Funding acquisition, Q.A.; Investigation, S.I., N.M. and I.Q.; Methodology, S.I., N.M., C.M.N.F. and A.H.; Project administration, H.A.; Resources, H.A., C.M.N.F., Q.A., I.Q. and A.H.; Software, S.I., C.M.N.F. and Q.A.; Supervision, H.A.; Validation, N.M., Q.A., I.Q. and A.H.; Visualization, H.A. and N.M.; Writing—original draft, S.I., H.A., N.M., C.M.N.F., Q.A. and A.H.; Writing—review and editing, Q.A. and I.Q. All authors have read and agreed to the published version of the manuscript.

**Funding:** This work was supported and funded by the Deanship of Scientific Research at Imam Mohammad Ibn Saud Islamic University (IMSIU) (grant number IMSIU-RP23075).

**Institutional Review Board Statement:** This study was conducted in accordance with the Declaration of Helsinki and approved by the Ethics Review Committee of National Textile University, Faisalabad 37610, Pakistan (Letter Number NTU/ERC/70).

**Informed Consent Statement:** Informed consent was obtained from all subjects involved in the study.

**Data Availability Statement:** The datasets generated during and/or analyzed during the current study are available from the corresponding author upon reasonable request.

**Acknowledgments:** This work was supported and funded by the Deanship of Scientific Research at Imam Mohammad Ibn Saud Islamic University (IMSIU) (grant number IMSIU-RP23075).

**Conflicts of Interest:** The authors declare no conflict of interest.

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
