# Peer review of "Recognition of Student Engagement State in a Classroom Environment Using Deep and Efficient Transfer Learning Algorithm"

_applsci, doi:10.3390/app13158637_

Round 1

Reviewer 1 Report

My opinions:

1.       The section Introduction is uselessly long. Author use in some part very old refrences.

2.       I propose that the authors divided or reduce section Introduction.

3.       Dataset of 45 students as tested sample is very small for good results of statistical evaluation.

4.       The Related work section is very well prepared.

5.       Did the authors get permission from the students to captured video?

6.       The consent was also confirmed by the so-called ethics commission?

7.       The authors state the accuracy in the paper. However, there are no graphs confirming this claim.

8.       Table 3 shows the years 2015 and 2017, where other researchers use the SVM method. However, the difference in accuracy is considerable. How do you explain the difference between 62% and 90%? Is it caused by a combination of SVM and logistic regression methods?

9.       VGG16 is VGG model of classical convolution nets. The paper has very good research quality but my question is, what new brings authors....

1.   In the paper me missed definition of research problem, research questions and hypothesis.

Quality is good.

Author Response

Original Manuscript ID:  ID: applsci-2507906 

Original Article Title: Recognition of Student Engagement State in a Classroom Environment using Deep and Efficient Transfer Learning Algorithm

To: Editor in Chief,

MDPI, Applied Sciences

Re: Response to reviewers

Dear Editor,

Many thanks for insightful comments and suggestions of the referees. Thank you for allowing a resubmission of our manuscript, with an opportunity to address the reviewers’ comments.

We are uploading (a) our point-by-point response to the comments (below) (response to reviewers), (b) an updated manuscript with yellow highlighting indicating changes, and (c) a clean updated manuscript without highlights (PDF main document).

By following reviewers’ comments, we made substantial modifications in our paper to improve its clarity, English and readability. In our revised paper, we represent the improved manuscript such as:

(1) Revised Abstract, (2) Revised Introduction, (3) Results section, (4) Discussions and Conclusion sections.

We have made the following modifications as desired by the reviewers:

Best regards,

Corresponding Author,

Dr. Qaisar Abbas (On behalf of authors),

Professor.

Reviewer 2 Report

In this paper the authors present a transfer learning model for measuring students' engagement state in a classroom. The subject of the article is interesting and worthy of discussion.

The motivation and research questions are described, and the main contributions of the work are identified. The authors also describe the methodology and present the results in some detail. Several experiments were also carried out and described to support the answers to the research questions.  The results are interesting.

The structure of the paper is adequate. However, it is necessary to review the numbering of headings (e.g., there is no section 3, the numbering of sections "5.1 Limitations" and "5.2 Future work", is not correct).

Figures are adequate.

References are appropriate.

It would be interesting if the authors could also include the accuracy rates of each solution in Table 1.

In table 1, in the field "Key features", the authors indicate "Predefine features of engagement and non-engagement" regarding the proposed method. They should consider being more specific, because in the other solutions specific features are presented.

(Line 439) The statement "Table 3 displays the evaluated and optimal hyperparameter values for the proposed method" does not seem to correspond to the content of the referred table.

(Line 476) There is a caption "Table 3 Comparison of accuracy with existing methods" which seems to be misplaced.

The authors state that "Both genders have participated voluntarily to conduct the underlying research" (4.1 Data Acquition) but make no reference to the existence of an opinion from an ethics committee. Since the study involves images of students, and their treatment to obtain information about their affective state, information should be included on whether there was an opinion from an ethics committee.

In table 3, the authors "... compared the accuracy of our proposed model with the existing methods". The results of the proposed approach are positive when compared to the other approaches, but it is important to consider that most of the methods used in the comparison used an annotation with 3 or more distinct states, while the proposed method only considers 2 states. This information should be considered in the table and should be included when analyzing the results and when making comparisons with other approaches.

Author Response

(The authors gave the same response as above.)

Round 2

Reviewer 1 Report

In this moment I have no requirements.

In this moment I have no requirements.

Reviewer 2 Report

In this revision, the authors have made an effort to address some of the concerns given to them in the review.

They have improved the information in some tables.

They improved some sections namely results, Discussions and Conclusion).

They updated some sentences and corrected some typos.

Also important, they included information about the informed consent that was obtained from all subjects involved in the study and about approval by the Ethics Review Committee.

Globally, the manuscript shows improvements over the version previously presented. I believe it was improved enough for meeting publication standards.